

# Multi level perspectives in stock price forecasting: ICE2DE-MDL

Zinnet Duygu Akşehir and Erdal Kılıç

Computer Engineering, Ondokuz Mayis University Samsun, Samsun, Turkey

## ABSTRACT

This study proposes a novel hybrid model, called ICE2DE-MDL, integrating secondary decomposition, entropy, machine and deep learning methods to predict a stock closing price. In this context, first of all, the noise contained in the financial time series was eliminated. A denoising method, which utilizes entropy and the two-level ICEEMDAN methodology, is suggested to achieve this. Subsequently, we applied many deep learning and machine learning methods, including long-short term memory (LSTM), LSTM-BN, gated recurrent unit (GRU), and SVR, to the IMFs obtained from the decomposition, classifying them as noiseless. Afterward, the best training method was determined for each IMF. Finally, the proposed model's forecast was obtained by hierarchically combining the prediction results of each IMF. The ICE2DE-MDL model was applied to eight stock market indices and three stock data sets, and the next day's closing price of these stock items was predicted. The results indicate that RMSE values ranged from 0.031 to 0.244, MAE values ranged from 0.026 to 0.144, MAPE values ranged from 0.128 to 0.594, and R-squared values ranged from 0.905 to 0.998 for stock indices and stock forecasts. Furthermore, comparisons were made with various hybrid models proposed within the scope of stock forecasting to evaluate the performance of the ICE2DE-MDL model. Upon comparison, The ICE2DE-MDL model demonstrated superior performance relative to existing models in the literature for both forecasting stock market indices and individual stocks. Additionally, to our knowledge, this study is the first to effectively eliminate noise in stock item data using the concepts of entropy and ICEEMDAN. It is also the second study to apply ICEEMDAN to a financial time series prediction problem.

## INTRODUCTION

Forecasting stock prices or market trends, positioned at the confluence of finance and computational technology, stands as a captivating subject of keen interest among both researchers and investors. The allure stems from the potential of proposed forecasting models to yield successful predictions, aiding investors in optimizing their investment returns. Nonetheless, the inherent complexity of the stock market, influenced by myriad unpredictable factors like corporate strategies, political developments, investor sentiments, and overall economic conditions, renders accurate prediction a challenging endeavor. In the literature, numerous forecasting models have been proposed within the scope

Corresponding author
Zinnet Duygu Akşehir,
duygu.aksehir@bil.omu.edu.tr

of a stock market index or stock price forecasting, in other words, financial time series forecasting. These models encompass a variety of techniques, including statistical methods like autoregressive integrated moving average (ARIMA) (*Babu & Reddy, 2015*) and generalized autoregressive conditional heteroscedastic (GARCH) (*Ariyo, Adewumi & Ayo, 2014*), machine learning methods such as support vector machine (SVM) (*Wen et al., 2010*), random forest (*Lin et al., 2017*), artificial neural networks (ANN) (*Altay & Satman, 2005*), and deep learning methods like long-short term memory (LSTM) (*Akşehir & Kılıç, 2019*), gated recurrent unit (GRU) (*Gupta, Bhattacharjee & Bishnu, 2022*), and convolutional neural network (CNN) (*Akşehir & Kiliç, 2022*). Although some success has been observed with prediction models utilizing statistical methods, their unsuitability for stock forecasting arises from the assumption that time series data is linear and stationary, these characteristics are not often present in financial time series. This occurs because these forecasting models presume linearity and stationarity within time series data, whereas the financial data utilized in this context demonstrate non-linear and non-stationary attributes. To address the constraints of statistical methods in stock forecasting, machine learning and deep learning-based models have been introduced. Nevertheless, despite their superiority over statistical methods, these approaches are susceptible to the intricate and dynamic characteristics of financial time series. Given the dynamic, non-stationary and nonlinear characteristics of financial market data, attaining dependable results proves challenging with a singular machine learning or deep learning approach for forecasting. Research has stated that these disadvantages can be overcome with a hybrid forecasting model, as opposed to the basic forecasting model, in which a single machine learning or deep learning method is used for stock forecasting (*Cui et al., 2023*; *Lv et al., 2022*; *Chopra & Sharma, 2021*; *Kanwal et al., 2022*; *Zhang, Sjarif & Ibrahim, 2022*).

## Literature review

In literature, studies within the scope of stock prediction have emphasized that the proposed hybrid models outperform basic models, and this has been proven with experimental results. Based on this inference, *Cui et al. (2023)* conducted research on the proposed hybrid models in the realm of stock forecasting, and as a result of this research, they grouped the hybrid models into three categories:

- *Pure machine/deep learning hybrid model:* This hybrid model combines deep learning methods with machine learning or other deep learning techniques. In this context, various stock forecasting models are included in the literature. *Chaudhari & Thakkar (2023)* introduced three feature selection approaches; top-m, k-means, and median range based on the coefficient of variation for stock and stock market index forecasting. Various deep learning and machine learning methods, such as back-propagation neural networks (BPNN), LSTM, GRU, and CNN, have been used to evaluate the proposed feature selection approaches. *Albahli et al. (2023)* proposed a forecasting model to predict the future trend of stocks and to help investors in the decision-making process based on this forecast. Stocks' historical price data were obtained for the suggested prediction model, and 18 technical indicator values were calculated from this data. Then, the auto-encoder method was applied to 18 technical indicators to decrease the

feature count and obtain a leaner set of technical indicators. The stocks' historical price data and the reduced technical indicator data were combined and given as input to the DenseNet-41 model, and the prediction was realized. In another study conducted by *Albahli et al. (2022)*, the same prediction framework was used, and the only difference between the two prediction frameworks is the use of the 1D DenseNet model instead of the DenseNet-41 model. In another study (*Yoo et al., 2021*) where the auto-encoder method was used, the DAE-LSTM hybrid prediction model consisting of denoising auto-encoder (DAE) and LSTM model was proposed to predict change points in stock market data. In the proposed model, after removing irrelevant features from financial market data with DAE, training was conducted using the LSTM model. *Chen, Wu & Wu (2022)* stated that most of the stock forecasting studies in the literature use a single stock data. It was emphasized that this causes the situations of influencing each other of the stocks in the similar group to be ignored. Accordingly, they proposed a new deep learning-based hybrid model to improve stock prediction performance. In the proposed KD-LSTM model, 16 bank stocks with similar trends listed on the Chinese stock market were clustered using the k-means method with a dynamic time warping (DTW) distance metric. According to the clustering results, four bank stock data, determined to belong to the same cluster each time, were used to train the LSTM model. *Rekha & Sabu (2022)* proposed a cooperative hybrid model combining deep learning methods, deep auto-encoder, and sentiment analysis for stock market prediction. In this proposed model, first of all, the noise in the stock data is effectively eliminated by the deep auto-encoder method. Then, sentiment analysis was performed on the news data related to the stock using VADER, and the sentiment index was obtained. The denoised historical price data and sentiment index were combined and given as input to the LSTM/GRU model for prediction. *Tao et al. (2022)* introduced a hybrid deep learning model focusing on the correlation between stocks and mutation points. This proposed hybrid model consists of three subnetworks, and different features about the stock were obtained in each subnetwork. Then, these attributes were combined for the prediction of the stock's closing price. *Polamuri, Srinivas & Mohan (2022)* stated that generative adversarial network (GAN) is not preferred in stock prediction because hyperparameter tuning is difficult. They considered this as a primary motivation and proposed a hybrid prediction framework using GAN to predict the stock market. This model used LSTM and CNN for the generator and discriminator, the two essential components of GAN, respectively. Additionally, Bayesian optimization and reinforcement learning were used to overcome the hyperparameter tuning difficulty of the GAN. In this model, an auto-encoder was used for feature extraction from stock market data, XGBoost was used for feature selection, and the PCA method was used for dimensionality reduction. *Zhou, Zhou & Wang (2022)* introduced a hybrid prediction model termed FS-CNN-BGRU, integrating feature selection, Bidirectional Gated Recurrent Unit (BiGRU), and CNN techniques for stock prediction. Another investigation introduced the BiCuDNNLSTM-1dCNN hybrid approach for stock price forecasting, employing Bidirectional Cuda Deep Neural Network Long Short Term Memory and 1-D CNN methodologies (*Kanwal et al., 2022*). The hybrid model detailed above enhances the accuracy of prediction results.

However, much like its counterpart, it is susceptible to constraints stemming from the sample-dependency issue.

- *Time series-deep learning hybrid model:* This hybrid model combines deep learning with traditional time series methods like ARIMA and GARCH. While there are numerous stock prediction models in the literature, there are relatively fewer studies on the time series-deep learning hybrid model compared to the hybrid models that fall into the other two categories. *Rather, Agarwal & Sastry (2015)* introduced a hybrid model that combines linear models, including autoregressive moving average (ARMA) and exponential smoothing, with a non-linear model, recurrent neural network (RNN), for predicting stock returns. The HAR-PSO-ESN model, introduced by *Ribeiro et al. (2021)*, integrates heterogeneous autoregressive (HAR) specifications with echo state networks (ESN) and particle swarm optimization (PSO) metaheuristic techniques to enhance the prediction accuracy of stock price return volatility. This type of hybrid prediction models addressing traditional time series methods allows for separating nonlinear and linear components within the data. Still, it is ineffective enough in filtering out excessive noise, such as unusual spikes and jumps.

- *Decomposition-deep learning hybrid model:* This type of hybrid model is obtained by combining deep learning methods with wavelet transform or decomposition approaches. Various stock market prediction models in the literature are based on this structure. Indeed, it has been observed that in these hybrid models, wavelet transform (*Qiu, Wang & Zhou, 2020*; *Bao, Yue & Rao, 2017*; *Tang et al., 2021*; *Wen et al., 2024*), Fourier transform (*Song, Baek & Kim, 2021*), and decomposition algorithms (*Cui et al., 2023*; *Lv et al., 2022*; *Liu et al., 2022a*; *Wang, Cheng & Dong, 2023*; *Yan & Aasma, 2020*; *Liu et al., 2022b*; *Rezaei, Faaljou & Mansourfar, 2021*; *Cao, Li & Li, 2019*; *Wang et al., 2022*; *Liu et al., 2024*; *Nasiri & Ebadzadeh, 2023*; *Yao, Zhang & Zhao, 2023*) are utilized with deep learning. *Tang et al. (2021)* applied the denoising approach consisting primarily of wavelet transform and singular spectrum analysis (SSA) on the financial time series to predict the Dow Jones Industrial Average Index (DJIA). Then, the LSTM model was trained on the denoised data. The Fourier transform employed in this context decomposes the time series into frequency components, but this transformation cannot describe time and frequency scale changes. Therefore, it is not effective in analyzing time-varying signals. To overcome this disadvantage of the Fourier transform, wavelet transform is also used with deep learning methods in stock forecasting models (*Qiu, Wang & Zhou, 2020*; *Bao, Yue & Rao, 2017*; *Tang et al., 2021*). Nevertheless, the efficacy of the wavelet transform hinges on parameter selection, including parameters like the count of layers and the selection of the fundamental wavelet function, thereby constraining the predictive model's performance. To overcome these disadvantages of Fourier and wavelet transform, decomposition approaches were proposed that decompose the time series into distinct frequency spectrums. In the proposed deep learning-based hybrid models, it was observed that decomposition approaches such as variational mode decomposition (VMD) (*Cui et al., 2023*; *Liu et al., 2022a*; *Wang, Cheng & Dong, 2023*; *Liu et al., 2024*; *Nasiri & Ebadzadeh, 2023*), empirical mode decomposition (EMD) (*Rezaei, Faaljou & Mansourfar, 2021*; *Yao, Zhang & Zhao, 2023*),

complete ensemble empirical mode decomposition (CEEMD) (*Yan & Aasma, 2020*; *Liu et al., 2022b*; *Rezaei, Faaljou & Mansourfar, 2021*), complete ensemble empirical mode decomposition with adaptive noise (CEEMDAN) (*Lv et al., 2022*; *Cao, Li & Li, 2019*) and improved CEEMDAN (ICEEMDAN) (*Wang et al., 2022*) were more preferred than wavelet and Fourier transform. The decomposition approaches used in stock prediction models enable a better understanding and modeling of the complex structure of financial time series. Therefore, this contributes to more accurate predictions of future values. However, the type of decomposition approach to be used in these prediction models also directly affects the model's prediction performance. After examining literature studies, it was noted that the VMD and CEEMDAN approaches were preferred more than other decomposition approaches. The utilization of these approaches demonstrated more successful results compared to both basic and other types of hybrid models. Additionally, ICEEMDAN, one of the mode decomposition approaches, was proven to achieve significant success in removing noise in time series data used in various fields, such as biomedical signal processing (*Colominas, Schlotthauer & Torres, 2014*), solar irradiance forecasting (*Sibtain et al., 2021*), and traffic flow prediction (*Gao, Jia & Yang, 2022*). WWhen analyzing the stock prediction models in existing literature, it was noted that despite this success of ICEEMDAN, it has only been used in one study (*Wang et al., 2022*).

Following the above analyses, this study presents a novel hybrid model named ICE2DE-MDL for predicting the closing value of the stock and stock market index. The proposed model, ICE2DE-MDL, comprises a two-level decomposition based on the ICEEMDAN algorithm, along with entropy and deep learning/machine learning methods. In the proposed ICE2DE-MDL model, techniques such as LSTM, LSTM-BN, GRU, and SVR are preferred for specific reasons:

- LSTM was chosen for its proficiency in capturing temporal dependencies in time series data.
- LSTM-BN was favored for its ability to expedite training and enhance generalization performance through batch normalization.
- GRU was selected because it can capture temporal dependencies similar to LSTM while utilizing fewer parameters.
- SVR was included for its effectiveness in handling complex relationships and mitigating overfitting.

Integrating these techniques aims to enhance the ICE2DE-MDL model's ability to cope with the complexities inherent in stock market prediction tasks.

## Motivation and contributions

This study addresses the challenges in predicting stock market index and stock closing prices, a crucial task for investors and financial institutions amidst the volatility of financial markets. Enhancing the accuracy of these predictions necessitates integrating not only traditional statistical methods but also novel techniques such as deep learning and machine

learning. In this context, this study aims to solve this challenge with the developed ICE2DE-MDL model.

The ICE2DE-MDL model amalgamates various innovative techniques to tackle the complexity inherent in financial time series data. Initially, the secondary decomposition of financial data is performed using the ICEEMDAN algorithm, facilitating data segregation into more meaningful and noise-free components. Subsequently, components are classified using entropy metrics like sample and approximate entropy, aiding in identifying high-frequency and noise-free components. Following this, different deep learning and machine learning models are separately trained for each component, ensuring better alignment with the unique characteristics of each component. This approach enables the model to predict fluctuations in financial data more effectively.

The contributions and innovations of this study are outlined as follows:

1. The literature review revealed the effectiveness of the ICEEMDAN method in eliminating noise from various time series. However, despite its successful application in diverse domains, including one study on financial time series data, the significance of applying ICEEMDAN to this specific data type remains noteworthy. Therefore, as far as we are aware, this study represents the second application of ICEEMDAN to a financial time series prediction problem, expanding the current understanding of its potential within this domain.

2. This study introduces an innovative denoising approach designed to remove noise in stock data. Recognizing the crucial role of noise reduction in enhancing the accuracy of stock market analyses, our proposed method utilizes ICEEMDAN-based secondary decomposition along with sample and approximate entropy. Previous forecast models lack a denoising approach, specifically incorporating both ICEEMDAN and entropy concepts. Therefore, to our knowledge, this study is the first to propose a denoising approach combining ICEEMDAN and entropy concepts, providing a unique and effective method for eliminating noise in stock market index and stock data. Our proposed denoising approach introduces a second decomposition step instead of directly eliminating high-frequency components from the dataset. This decision is based on the recognition that these components may contain valuable information, and a secondary decomposition allows for a more nuanced and effective noise elimination process.

3. Previous studies proposing decomposition based on deep learning hybrid models usually train the IMFs or subseries, obtained from decomposition, with the same model. However, since each IMF possesses distinct characteristics, it is imperative to identify and train the most suitable model for each IMF individually rather than applying a uniform model for all intrinsic mode functions (IMFs). The noiseless IMFs were trained with four different models: LSTM, GRU, LSTM with batch normalization (LSTM-BN), and support vector regression (SVR). The model achieving the lowest error metric among these options was subsequently chosen as the optimal prediction model for the corresponding IMF.

4. Our proposed ICE2DE-MDL model has demonstrated superior performance to both fundamental models and decomposition based on deep learning hybrid models, as

evidenced by compelling experimental results. The enhanced accuracy and predictive capability of the ICE2DE-MDL showcase its effectiveness in addressing the challenges posed by existing models, marking a notable advancement in predictive modeling for specific applications or domains.

### Organization

The subsequent sections of this paper are organized as follows: The "Related Methodology" section discusses the methodologies. In the section titled "The ICE2DE-MDL Prediction Model," the framework of the proposed hybrid model is outlined. Details regarding the dataset utilized, hyperparameter settings of the model, selected benchmark models, and performance evaluation metrics are provided in the "Experimental Settings" section. The "Results and Discussion" section comprehensively analyzes the experimental results obtained in this investigation. Ultimately, the conclusion and recommendations for future research are delineated in the concluding section.

## RELATED METHODOLOGY

This study introduces the ICE2DE-MDL prediction model, a novel framework that combines ICEEMDAN-based secondary decomposition, entropy, and LSTM, GRU, LSTM-BN, and SVR models to enhance the accuracy of stock index predictions. The subsequent section outlines the fundamental principles behind each method integrated into the proposed ICE2DE-MDL model.

### Improved complete ensemble empirical mode decomposition with adaptive noise

The Empirical Mode Decomposition (EMD) technique, pioneered by *Huang et al. (1998)*, dissects a time series into IMFs characterized by distinct frequencies and scales. Nonetheless, EMD is susceptible to 'mode mixing,' where similar oscillations may manifest across distinct modes or exhibit varying amplitudes within a single mode. To address this limitation, Ensemble Empirical Mode Decomposition (EEMD) was introduced as a solution (*Wu & Huang, 2009*). This method describes the actual IMF components by taking the average of multiple trials. It is also an approach that adds white noise to each trial, thus enabling cleaner/less noisy IMFs to be obtained compared to EMD. Despite these advantages, EEMD has challenges, like the high computational burden and inadequate elimination of white noise. In light of these challenges, *Torres et al. (2011)* presented the CEEMDAN technique, aiming to rectify the deficiencies of EEMD and provide a more efficient decomposition approach. In addition to effectively eliminating the mode mixing issue, CEEMDAN presents the benefits of minimal reconstruction error and substantially decreased computational cost. The CEEMDAN marks a notable advancement over EEMD by achieving minimal reconstruction error and resolving the challenge of variable mode counts across different signal-plus-noise realizations. Nonetheless, CEEMDAN still exhibits certain aspects requiring enhancement: (i) its modes retain some residual noise, and (ii) the signal information seems "delayed" compared to EEMD, manifesting some

---

**Algorithm 1** ICEEMDAN Decomposition Method

---

Output: $\widetilde{IMF_k}$

$residue_t \leftarrow$ Residues

$E_t(.) \leftarrow t$-th IMF extracted through EMD

$M(.) \leftarrow$ Mean value of the IMF

$wn^{(i)} \leftarrow$ White noise

$x \leftarrow$ Time series signal

$\beta \leftarrow$ Level of noise

$E_t(wn^{(i)}) \leftarrow t$-th EMD component of the $wn^{(i)}$;

**Stage 1: Obtain the signal to be decomposed and calculate the first residue**

$$x^{(i)} = x + \beta_0 E_1(wn^{(i)})$$

$$residue_1 = M(x^{(i)})$$

**Stage 2: Calculate of the first IMF ($t = 1$)**

$$\widetilde{IMF_1} = x - residue_1$$

**Stage 3: Obtain the second IMF**

$$\widetilde{IMF_2} = residue_1 - residue_2 = residue_1 - M(residue_1 + \beta_1 E_2(wn^{(i)}))$$

**Stage 4: For $t = 3, \ldots,$ T calculate the $t$th residue**

$$residue_t = M(residue_{t-1} + \beta_{t-1} E_t(wn^{(i)}))$$

**Stage 5: Calculate the $t$th IMF**

$$\widetilde{IMF_t} = residue_{t-1} - residue_t$$

**Stage 6: Proceed to Stage 4 for the subsequent $t$**

---

"spurious" modes during the early phases of decomposition. To further enhance the method, _Colominas, Schlotthauer & Torres (2014)_ introduced an improved version known as Improved CEEMDAN (ICEEMDAN), which mitigates the residual noise issue and addresses the 'delayed' signal problem. The fundamental steps of ICEEMDAN are outlined in Algorithm 1.

## Entropy

Entropy, grounded in information theory principles, serves as a metric to quantify the intricacy or irregularity observed within a time series. Understanding the unpredictability and speculative nature of price movements in financial markets is crucial. Although various entropy metrics exist in the literature, two widely utilized types are sample entropy (_Richman & Moorman, 2000_) and approximate entropy (_Pincus, 1991_). Approximate entropy quantifies the likelihood that similar patterns of observations will persist unchanged over consecutive data points. In contrast, sample entropy quantifies the regularity or predictability of fluctuations in a time series. The computation of these metrics is outlined in Algorithm 2. In financial time series, elevated entropy values may signal speculative and unpredictable price movements. These higher entropy levels suggest increased complexity and irregularity in market dynamics, potentially prompting investors to take risks. However, such unpredictability also carries the risk of financial losses. Consequently, assessing a financial time series's entropy value becomes crucial when formulating a robust investment strategy.

## Long short term memory and long short term memory with batch normalization

Long short-term memory is a variant of recurrent neural network (RNN) architecture tailored to address the vanishing gradient issue commonly experienced by conventional RNNs, especially when processing extensive sequential data. While both LSTM and traditional RNNs share the characteristic of having repeating modules, their architectures differ significantly. The primary distinction lies in handling sequential information and mitigating the vanishing gradient problem.

In a standard RNN, a single layer processes input sequences sequentially using a hyperbolic tangent (tanh) activation function. However, the challenge of capturing long-range dependencies effectively persists due to the vanishing gradient problem. In contrast, the LSTM architecture (see Fig. 1) tackles this issue by introducing a more intricate structure. This complex structure comprises four interconnected layers within each repeating module: the cell state, output gate, input gate, and forget gate. These components work in tandem to selectively retain or discard information, update the cell state, and produce the output for each time step. This intricate design allows LSTMs to

**Algorithm 2** Compute the Sample and Approximate Entropy

Input: *Time series data*

Output: *SampleEntropy*, *ApproximateEntropy*

$x(k), k = 1, 2, .., L \leftarrow$ A time series

$d \leftarrow$ Embedding dimension

$r \leftarrow$ Tolerance value

**Stage 1: Extend $x(k)$ to the $d^{th}$ vector $U_d(k)$**

$U_d(k) = [x(k), u(k+1), ..., u(k+d-1)] \leftarrow k = 1, 2, ..., L-d+1$

**Stage 2: Calculate the distance between $U_d(k)$ and $U_d(j)$**

$D[U_d(k), U_d(j)] = max_{t=0,1,...,d-1}\{|x(k+t) - x(j+t)|\} \leftarrow j = 1, 2, ..., L-d+1$ and $j \neq k$

**Stage 3: Calculate the approximate entropy**

- *Assess the frequency of patterns and regularity within the specified tolerance $r$:*

$$C_k^d(r) = \frac{\text{Number of j such that } D[U_d(k), U_d(j)] \leq r}{L-d+1}$$

- Calculate the average of the logarithm of $C_k^d(r)$:

$$\psi^d(r) = \frac{\sum_{k=1}^{L-d+1} ln[C_l^d(r)]}{L-d+1}$$

- The ApproximateEntropy can be characterized as:

$$ApproximateEntropy(d, r) = \psi^d(r) - \psi^{d+1}(r)$$

**Stage 4: Compute sample entropy**

- Calculate the pair of coefficients and determine their sum:

$$A_k^d(r) = \frac{\sum_{j=1,j\neq k}^{L-d} \text{number of times that} D[U_{d+1}(k), U_{d+1}(j)] < r}{L-d-1}$$

$$B_k^d(r) = \frac{\sum_{j=1,j\neq k}^{L-d} \text{number of times that} D[U_d(k), U_d(j)] < r}{L-d-1}$$

$$A^d(r) = \frac{\sum_{k=1}^{L-d} A_k^d(r)}{L-d}$$

$$B^d(r) = \frac{\sum_{k=1}^{L-d} B_k^d(r)}{L-d}$$

- The SampleEntropy can be characterized as:

$$SamppleEntropy(d, r) = -ln[\frac{A^d(r)}{B^d(r)}]$$

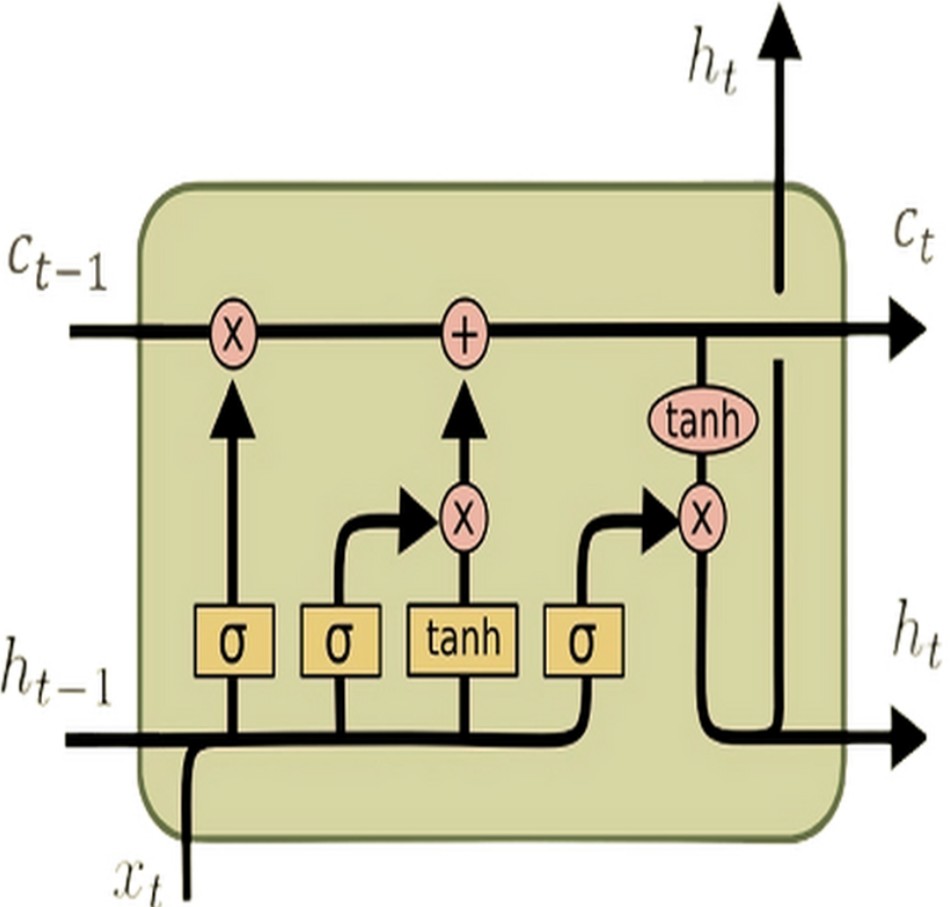

**Figure 1** **LSTM architecture.**

capture and remember long-term dependencies in sequential data more effectively than traditional RNNs.

The LSTM transaction equations are given as follows (*Hochreiter & Schmidhuber, 1997*):

$$forget_l = \sigma(W_{forget} \cdot [hidden_{l-1}, X_l] + b_{forget}) \tag{1}$$

$$input_l = \sigma(W_{input} \cdot [hidden_{l-1}, X_l] + b_{input}) \tag{2}$$

$$output_l = \sigma(W_{output} \cdot [hidden_{l-1}, X_l] + b_{output}) \tag{3}$$

$$\tilde{Cell}_l = tanh(W_{cell} \cdot [hidden_{l-1}, X_l] + b_{cell}) \tag{4}$$

$$Cell_l = forget_l \odot Cell_{l-1} + input_l \odot \tilde{Cell}_l \tag{5}$$

$$hidden_l = output_l \odot tanh(Cell_l). \tag{6}$$

The equations above outline the functionality of an LSTM unit at time step $l$ for an input vector ($X$). Here, $input_l$ represents an input gate, $forget_l$ denotes a forget gate, $output_l$ signifies an output gate, $Cell_l$ represents a memory cell, $hidden_l$ symbolizes a hidden state, $W$ denotes the weight matrix, $b$ indicates the bias vector, and $\sigma$ represents the activation function. The default connections between these units are illustrated in Fig. 1.

The long short-term memory with batch normalization (LSTM-BN), proposed by *Fang et al. (2023)* for predicting financial time series movement, enhances the conventional LSTM architecture by incorporating a batch normalization (BN) layer. Incorporating the batch normalization layer into this architecture is intended to improve the network's performance. The batch normalization layer makes the network faster and more stable during training.

## Gated recurrent unit

The gated recurrent unit (GRU) (*Cho et al., 2014*) is a form of RNNs engineered to tackle the vanishing gradient issue and capture long-range dependencies within sequential data, akin to LSTM. The GRU, illustrated in Fig. 2, is recognized for its simplified architecture compared to LSTM. It is achieved by combining the memory cell and hidden state into a single state, thus reducing the number of parameters.

The GRU comprises the update gate ($update_l$) and the reset gate ($reset_l$). These gates manage the flow of information within the unit, enabling selective updates to its hidden state. The following equations define the computations within a GRU:

$$update_l = \sigma(W_{update} \cdot [hidden_{l-1}, X_l] + b_{update}) \tag{7}$$

$$reset_l = \sigma(W_{reset} \cdot [hidden_{l-1}, X_l] + b_{reset}) \tag{8}$$

$$\tilde{hidden}_l = tanh(W_{hidden} \cdot [reset_l \odot hidden_{l-1}, X_l] + b_{hidden}) \tag{9}$$

$$hidden_l = (1 - update_l) \odot hidden_{l-1} + update_l \odot \tilde{hidden}_l. \tag{10}$$

As illustrated in the equations presented, $\sigma$ represents the sigmoid activation function, while tanh signifies the hyperbolic tangent activation function. The operator $\odot$ signifies element-wise multiplication. Moreover, $[hidden_{l-1}, X_l]$ represents the concatenation of the current input $X_l$ and previous hidden state $hidden_{l-1}$. $W$ denotes the weight matrix, and $b$ signifies the bias vector.

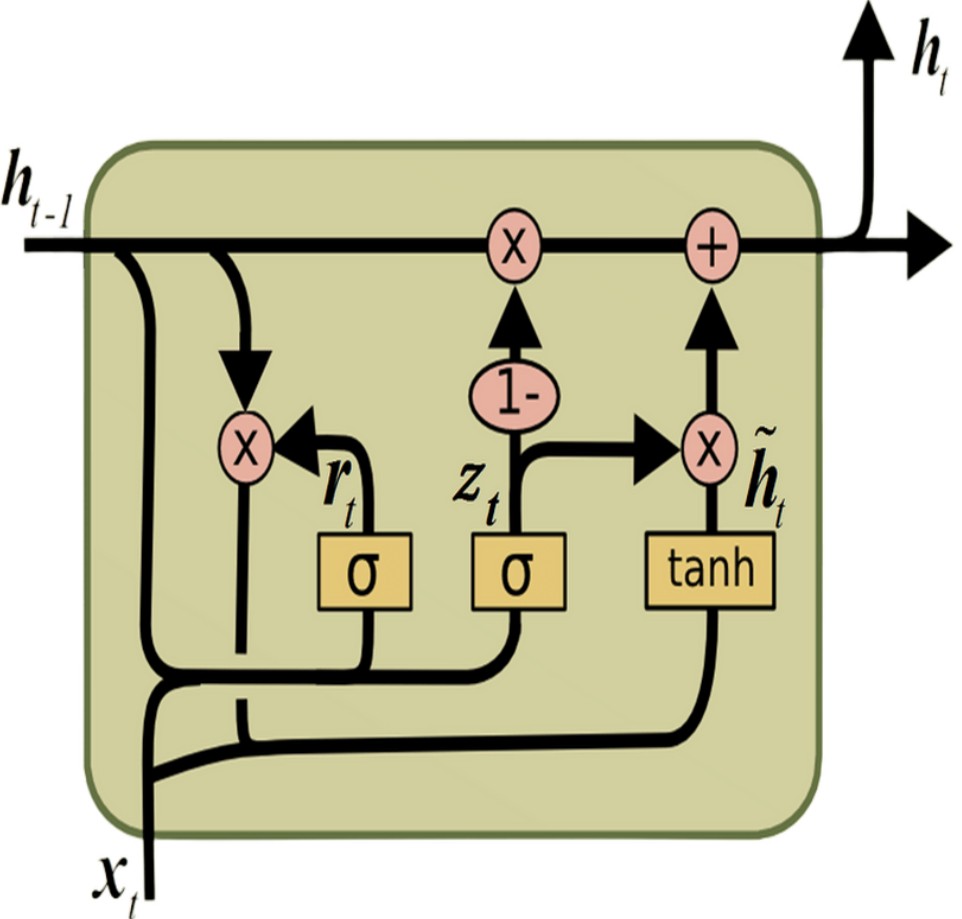

**Figure 2  GRU architecture.**

## Support vector regression

Support vector regression (*Cortes & Vapnik, 1995*), mainly employed for solving regression problems, belongs to the support vector machines (SVM) family. SVR aims to create a hyperplane and optimally fit data points onto this plane. Like classification problems, SVR's success in regression tasks is grounded in the concept of margin the distance between data points and the hyperplane. SVR enhances its generalization ability by maximizing this margin. The main parameters of SVR are as follows:

- *Kernel type:* Specifies the kernel function used to transform data points in a high-dimensional space. Different kernel types include polynomial, radial basis function (RBF), and linear kernels.
- *C (cost):* It is a hyperparameter that controls the width of the margin. Larger values of C result in a narrower margin but can also increase the model's tendency to overfit.
- *Epsilon ($\epsilon$):* Another hyperparameter that controls the width of the margin. Larger $\epsilon$ values result in a wider margin.

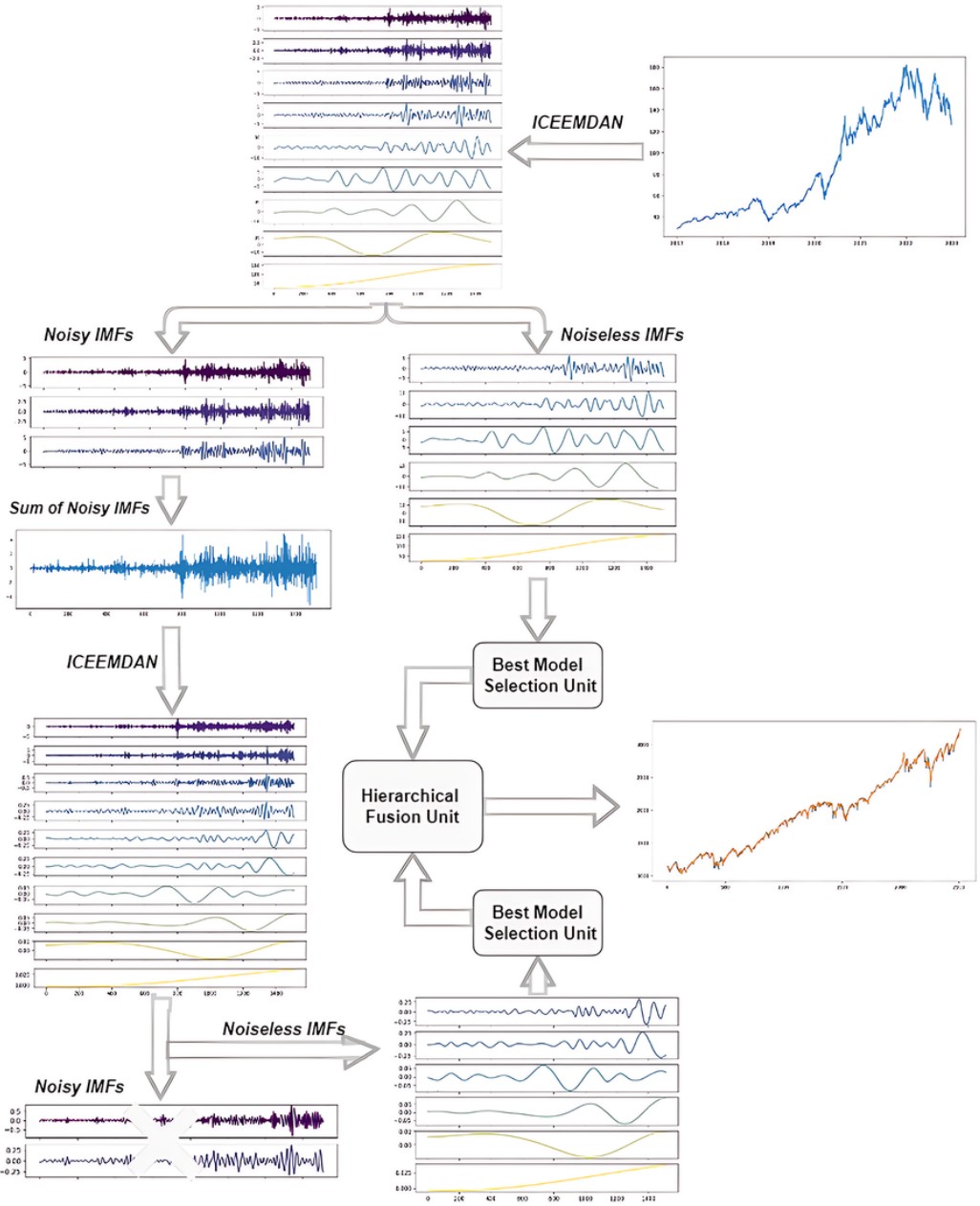

**Figure 3** The flowchart of the proposed ICE2DE-MDL model.

## THE ICE2DE-MDL PREDICTION MODEL

The study presents a novel hybrid model, termed ICE2DE-MDL, which is tailored for the prediction of closing prices in both stock market indices and individual stocks. The architectural layout of the model is visually shown in Fig. 3. Following that, a detailed breakdown of the model's intricacies is outlined, delineating each step comprehensively:

---

**Algorithm 3** ICEEMDAN and Entropy-Based Denoising Method

Input: *Time series data*

$Imfs : [Imf_0, Imf_1, \ldots, Imf_{l-1}] \leftarrow l$ IMFs derived through ICEEMDAN()

**procedure** SELECTION_IMF(*Imfs*)

    *list_entropy_sample* = []

    *list_entropy_approximate* = []

    **for** imf **in** Imfs **do**

        *list_entropy_sample.append(SampEn(imf))*

        *ist_entropy_approximate.append(ApEn(imf))*

        *total_SampEn = sum(list_entropy_sample)*

        *total_ApEn = sum(ist_entropy_approximate)*

        *ratio_sample_entropy* = $[\frac{entropy*100}{total\_SampEn}$ for entropy in list_entropy_sample]

        *ratio_approximate_entropy* = $[\frac{entropy*100}{total\_ApEn}$ for entropy in

ist_entropy_approximate]

    **end for**

    *high_frequency_imfs* = []

    *noiseless_imfs* = []

    **for** i **in** range(len(IMFs)) **do**

        **if** *ratio_sample_entropy*[i] > 20 OR *ratio_approximate_entropy*[i] > 20 **then**

            *high_frequency_imfs.append(IMFs[i])*

        **else**

            *noiseless_imfs.append(IMFs[i])*

        **end if**

    **end for**

    **return** *high_frequency_imfs*, *noiseless_imfs*

**end procedure**

*First_IMFs = ICEEMDAN(Input)*

*First_decomp_IMF_high_frequency, First_decomp_IMF_noiseless = Selection_IMF(First_IMFs)*

*high_frequency_First_IMFs_sum = sum(First_decomp_IMF_high_frequency)*

*Second_IMFs = ICEEMDAN(high_frequency_First_IMFs_sum)*

*Second_decomp_IMF_high_frequency, Second_decomp_IMF_noiseless = Selection_IMF(Second_IMFs)*

---

1.  Firstly, stock market index/stock closing prices are decomposed into IMFs by ICEEMDAN. Then, the sample and approximate entropy values of the IMFs are calculated. The proportional sample and approximate entropy values of each IMF are obtained using these calculated values. These entropy rates are used to classify IMFs as high-frequency and noiseless. For this, the entropy ratios are compared with a pre-defined threshold value. IMFs with an entropy ratio higher than the threshold value are classified as high-frequency, while others are classified as noiseless. While the

noiseless IMFs are used in the training, the high-frequency IMFs are forwarded to the next stage, the second decomposition.

2.  Contrary to the direct discarding of IMFs, which are determined as high-frequency in the first step, a second decomposition process is applied, considering their possibility of containing useful information. For this purpose, firstly, IMFs determined as high-frequency are summed, and the ICEEMDAN method is performed on the obtained time series. The IMFs acquired from the secondary decomposition are categorized as high-frequency and noiseless, as in the first step. The high-frequency components determined as a result of this classification are discarded, while the noiseless components are used for training. In this regard, the noise in this data has been effectively eliminated by applying the first two steps on the financial time series. The algorithmic flow of this proposed ICEEMDAN and entropy-based denoising method is given in Algorithm 3.

3.  After removing the noise, training is carried out separately on the noiseless IMFs obtained from both decompositions. Since the characteristics of each IMF are different, the best training model for them has been tried to be determined, rather than training them with a uniform model. For this purpose, each IMF is trained with four different models: LSTM, LSTM-BN, GRU, and SVR. The error metrics are compared, and the model exhibiting the lowest error metric is determined to be the best training model for that IMF. This process is performed at the Best Model Selection Unit in the framework provided in Fig. 3.

4.  After training each noiseless IMF with its best model, the prediction results obtained from these IMFs are combined hierarchically. In this hierarchical combining approach, the prediction results of the noiseless IMFs derived from the second decomposition are initially combined, followed by the fusion of this result with the prediction outcomes of the noiseless/noise-free IMFs derived from the initial decomposition. Thus, the final prediction result of the ICE2DE-MDL model is obtained. This process is performed at the hierarchical unit in the framework provided in Fig. 3.

## EXPERIMENTAL SETTINGS

The proposed ICE2DE-MDL model was employed to forecast the next day's closing prices for both stock market indices and individual stocks. Various experiments were then conducted on a computer equipped with an Intel i9-12900K processor to evaluate the predictive capabilities of the ICE2DE-MDL model. Initially, the proposed denoising method was applied to remove noise from the closing values of both indices and individual stocks. Subsequently, the prediction process utilized the resulting noiseless components. Within this section, we introduced the stock market index and stock datasets used in the study, followed by a discussion on the hyperparameter settings of the proposed ICE2DE-MDL model. Finally, the section concluded with details regarding the statistical metrics employed to evaluate the prediction model.

### Dataset

The prediction model's performance was evaluated and compared with current studies in the literature using 11 datasets comprising stock market indices and individual stocks. The

**Table 1  Statistical description of closing prices in stock market indices.**

| Stock market index | Count | Mean | Maximum | Minimum | Standard deviation |
|---|---|---|---|---|---|
| CAC 40 | 2,745 | 4,456.96 | 6,111.24 | 2,781.68 | 759.63 |
| FTSE 100 | 2,715 | 6,499.36 | 7,877.50 | 4,805.80 | 715.13 |
| KOSPI | 2,650 | 2,040.40 | 2,598.19 | 1,457.64 | 194.29 |
| NASDAQ 100 | 2,705 | 4,691.97 | 1,420.54 | 1,728.34 | 2,296.04 |
| NIKKEI | 2,628 | 16,198.27 | 24,270.62 | 8,160.01 | 5,022.47 |
| SET | 2,620 | 1,384.49 | 1,838.96 | 685.89 | 268.41 |
| S&P 500 | 2,705 | 2,042.35 | 3,580.84 | 1,022.58 | 642.33 |
| SSE | 2,610 | 2,817.38 | 5,166.35 | 1,950.01 | 519.32 |

**Table 2  Statistical description of stock closing price.**

| Stock | Count | Mean | Maximum | Minimum | Standard deviation |
|---|---|---|---|---|---|
| Xinning logistics | 2,613 | 7.49 | 30.75 | 2.23 | 3.96 |
| Zhongke electric | 2,610 | 5.40 | 11.49 | 1.96 | 1.94 |
| CITIC securities | 2,613 | 17.05 | 37.82 | 9.29 | 5.68 |

selected stock market indices included the Standard and Poor's 500 Index (S&P 500), France CAC 40 Index, Tokyo NIKKEI Index (Japan), Shanghai Stock Exchange Composite Index (SSE), SET Index (Thailand), FTSE 100 Index (London), Seoul KOSPI Index (Korea), and NASDAQ 100 Index (USA). Additionally, the study considered individual stocks from China's A-share market, specifically Xinning Logistics, Zhongke Electric, and CITIC Securities. This diversified dataset aims to comprehensively assess the model's predictive capabilities across various financial instruments and markets. The daily closing prices of stocks and stock market indices from January 1, 2010, to January 1, 2020, were gathered from the website https://finance.yahoo.com/.

The statistical analysis outcomes, comprising data quantity, minimum and maximum values, mean, and standard deviation for each dataset concerning both stocks and stock indices, were presented in Tables 1–2. When the statistical analysis results were examined, it was observed that there was a substantial gap between the maximum and minimum values, as well as high standard deviation values for both the stock market index and stock datasets. This observation suggests that the chosen stock market indices and individual stocks demonstrate considerable volatility and possess non-stationary characteristics.

## Hyperparameter settings of the proposed model

The hyperparameter values for the ICE2DE-MDL forecasting model were meticulously determined through a series of experiments, and the resulting values are detailed in Table 3. Accordingly, for the LSTM training model, three LSTM layers were utilized, each containing 128, 64, and 16 neurons, respectively. Between these LSTM layers, a Dropout layer with a ratio of 0.1 was included.

For the LSTM-BN network, a structure with two LSTM layers including 128 and 32 neurons, respectively, was adopted. After each LSTM layer, a batch normalization layer

**Table 3  Hyperparameters of the ICE2DE-MDL forecasting model.**

| Method | Hyperparameter | Value |
|---|---|---|
| **LSTM** | Time step | 10 |
| | Unit count | 128, 64, 16 |
| | Maximum epoch count | 200 |
| | Hidden layer count | 3 |
| | Activation function | ReLU |
| | Regularization | Early stopping |
| | Optimizer | Adam |
| | Dropout rate | 0.1 |
| | Loss function | Mean square error |
| **LSTM-BN** | Time step | 10 |
| | Unit count | 128, 32 |
| | Maximum epoch count | 200 |
| | Hidden layer count | 2 |
| | Activation function | ReLU |
| | Number of BN layer | 2 |
| | Optimizer | Adam |
| | Loss function | Mean square error |
| | Regularization | Early stopping |
| | Dropout rate | 0.1 |
| **GRU** | Time step | 10 |
| | Unit count | 128, 64, 32, 8 |
| | Maximum epoch count | 200 |
| | Hidden layer count | 4 |
| | Activation function | tanh |
| | Regularization | Early stopping |
| | Optimizer | Adam |
| | Dropout rate | 0.2 |
| | Loss function | Mean square error |
| **SVR** | Kernel function | Linear |
| | Cost | 10 |
| | Epsilon | 0.001 |
| | Cross-validation | 5-fold |
| **Approximate and sample entropy** | Embedding dimension | 2 |
| | Tolerance | 0.2 |

was incorporated, and a Dropout layer with a ratio of 0.1 was introduced following the final batch normalization layer.

The training model based on GRU architecture consisted of four layers with varying neuron counts: 128, 64, 32, and 8 neurons, sequentially. Following each GRU layer, a Dropout layer with a ratio of 0.2 was incorporated. Furthermore, across the LSTM, LSTM-BN, and GRU training models, the Adam optimizer was selected, the mean square error was chosen as the loss function, and the time step value was set to 10. Activation

functions included ReLU for the LSTM and LSTM-BN models and tanh for the GRU model.

In the SVR model, the kernel function was selected as linear. For the cost and epsilon parameters, 10 and 0.001 were used, respectively. In addition, 5-fold cross-validation was performed. The maximum epoch number was determined to be 200 for the four models used in the training, and early stopping was implemented to mitigate potential overfitting issues during training. The noise-free IMFs used in the training of the models were scaled with a standard scaler, and the training-test dataset was obtained by dividing each IMF dataset by 75–25%.

In calculating sample and approximate entropy values for the obtained IMFs, the embedding dimension and tolerance values were set to 2 and 0.2, respectively. To classify the obtained IMFs as either high-frequency or noiseless, a threshold for the entropy ratio was established at 20%. IMFs with approximate or sample entropy ratios exceeding 20% were designated high-frequency, while those below the threshold were categorized as noiseless.

## Benchmark models selection

Four different prediction models were selected to compare the performance of the proposed ICE2DE-MDL forecasting model with current studies in the literature. Details about the selected models are given below:

- *VMD-SE-GBDT-BiGRU-XGBoost (Wang, Cheng & Dong, 2023):* The proposed hybrid model, integrating VMD, sample entropy (SE), Gradient Boosting Decision Tree (GBDT), Bidirectional Gated Recurrent Unit (BiGRU), and XGBoost methods, was introduced for stock market index prediction. In this model, the index data is initially decomposed into subseries using VMD, followed by the calculation of sample entropy for each subseries. Subseries with similar sample entropy values were grouped and restructured. Next, the most influential price data and technical indicators affecting the index were determined using the GBDT method. The restructured subseries and the features identified by GBDT were then fed into the BiGRU model for prediction. The prediction results obtained from the BiGRU model were combined using the XGBoost method, and the final prediction result was obtained. The proposed VMD-SE-GBDT-BiGRU-XGBoost model was applied to the CSI 500, NASDAQ 100, FTSE 100, and France CAC 40 indices.

- *IVMD-ICEEMDAN-ALSTM (Wang et al., 2022):* The hybrid prediction model, consisting of secondary decomposition, multi-factor analysis, and attention-based LSTM (ALSTM), was proposed to forecast the stocks' closing prices. In the proposed model, two decomposition approaches, ICEEMDAN and VMD, were employed to eliminate noise in the financial time series. The ALSTM method was used as the training model. The proposed hybrid model was applied to the SSE, NIKKEI, KOSPI, and SET indices.

- *MS-SSA-LSTM (Mu et al., 2023):* The MS-SSA-LSTM hybrid model was proposed for predicting stock closing prices by integrating two data types. In this proposed hybrid model, comments related to stocks were initially gathered, and sentiment analysis was

**Table 4  The statistical metrics used in the analysis of prediction results.**

| Metrics | Descriptions | Equation |
|---|---|---|
| RMSE | Root Mean Square Error | $\sqrt{\frac{1}{n}\sum_{i=1}^{n}(P_i - A_i)^2}$ |
| MAE | Mean Absolute Error | $\frac{1}{n}\sum_{i=1}^{n}|P_i - A_i|$ |
| MAPE | Mean Absolute Percentage Error | $\frac{1}{n}\sum_{i=1}^{n}|\frac{P_i - A_i}{A_i}|$ |
| $R^2$ | R-square | $1 - \frac{\sum_{i=1}^{n}(P_i - A_i)^2}{\sum_{i=1}^{n}(P_i - \widetilde{A}_i)^2}$ |

performed to obtain a sentiment score. Subsequently, stocks' historical price data and sentiment scores were combined and given as input to the LSTM model optimized by the Sparrow Search Algorithm (SSA). The proposed MS-SSA-LSTM model was applied to six stocks in the Chinese market, including PetroChina, CITIC Securities, Guizhou Bailing, HiFuture Technology, Xinning Logistics, and Zhongke.

• *P-FTD-RNN/LSTM/GRU (Song, Baek & Kim, 2021):* The proposed hybrid forecasting model combined padding-based Fourier transformation with deep learning methods to predict the stock market index. This model uses padding-based Fourier transformation to reduce noise in financial time series data. Following this, denoised data was used to train GRU, RNN, and LSTM models. The proposed model was applied to S&P 500, SSE, and KOSPI index data.

## Evaluation metrics

We evaluated the proposed forecasting model using commonly employed metrics in the literature. The metrics calculations were defined in Table 4. Within these mathematical expressions, $n$ stands for the quantity of data points, $P$ symbolizes the predicted value, $A$ denotes the actual value, and $\widetilde{A}$ represents the average of the actual values.

## RESULTS AND DISCUSSION

In this section, we examined the outcomes of experiments conducted on different stocks and stock market indices to evaluate the predictive performance of the proposed ICE2DE-MDL model. Due to the page limit, it is impossible to show the proposed forecast model's detailed results on all stock market indices and stocks considered in this study. For this purpose, the KOSPI index dataset was selected as an example to detail the stages of the ICE2DE-MDL model.

As a first example, the denoising approach given in Algorithm 3was applied to the closing prices of the KOSPI index. In this process, 10 IMFs were obtained using the ICEEMDAN method as a result of this first decomposition. Figure 4 shows the KOSPI indices' closing value and the IMFs obtained from the first decomposition. After calculating the sample and approximate entropy ratios for these IMFs, the first two IMFs, which exhibited ratios surpassing a predefined threshold, were identified as high-frequency components. After summing the first two IMFs identified as high-frequency components, the process continued by applying the ICEEMDAN method for a second decomposition. Following this second decomposition, 10 IMFs were obtained. Evaluating the entropy ratios, it

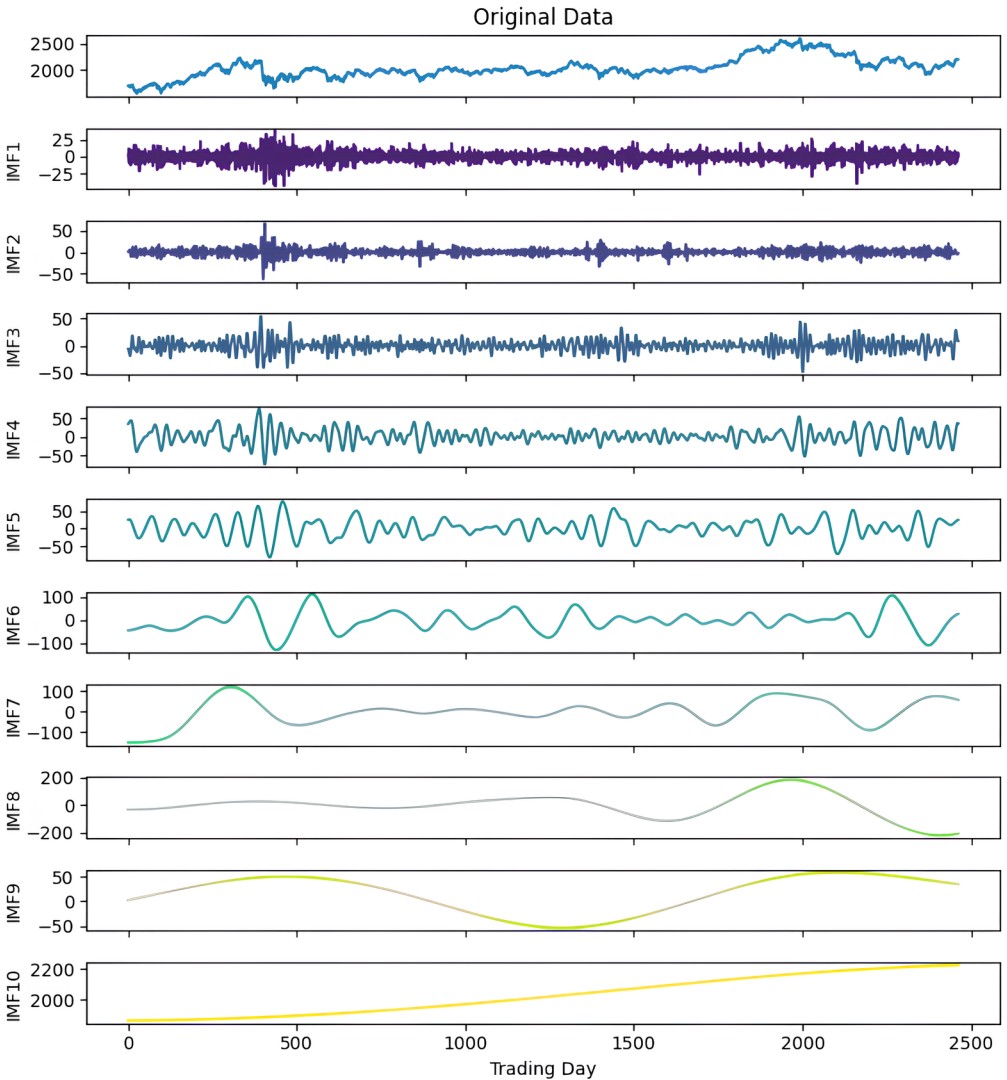

**Figure 4** **First decomposition results of KOSPI closing price.**

was determined that the first four IMFs from this set were classified as high-frequency components. The summation of the high-frequency components identified from the first decomposition and the IMFs obtained from the second decomposition were illustrated in Fig. 5. Furthermore, Table 5 includes the entropy values and ratios of the IMFs obtained from both the first and second decompositions ($Imf - i^{(j)}$ means the i-th IMF as a result of the j-th decomposition). The values highlighted in bold in the table represent the IMF components classified as high-frequency.

After completing the first and second decomposition steps, the denoised IMFs were fed into LSTM, LSTM-BN, GRU, and SVR models for training. Initially, each denoised IMF obtained from the two decomposition steps underwent individual training using these four methods. The model with the lowest error metrics, such as mean squared error, was

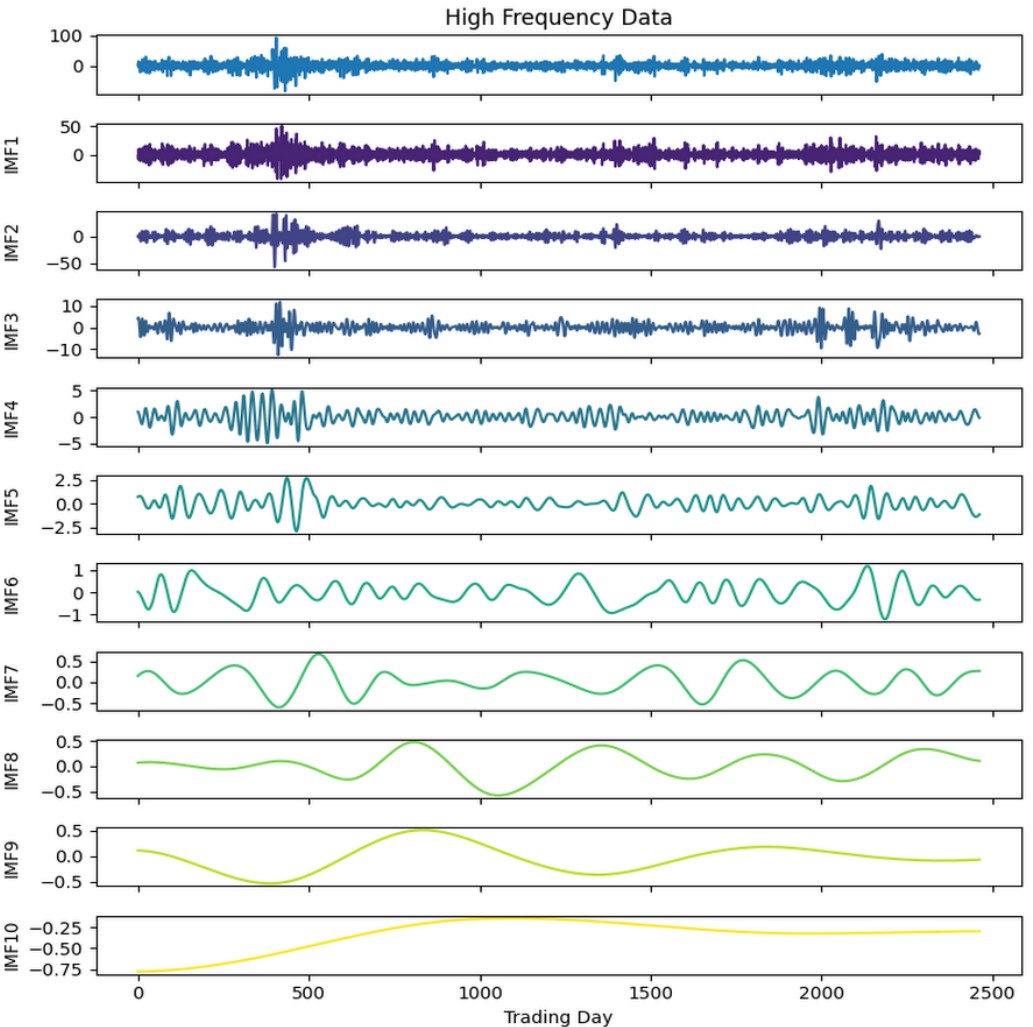

**Figure 5** Second decomposition results of KOSPI closing price.

selected as the best model for each respective IMF. Subsequently, the best prediction model results for the noiseless IMFs from the first and second decompositions were hierarchically combined, as outlined in Section 'The ICE2DE-MDL Prediction Model'.

All these procedures were similarly carried out for the remaining seven stock market indices and three individual stocks. Consequently, the prediction outcomes of the ICE2DE-MDL model for the stock market indices and individual stocks considered in this study are presented in Tables 6–7, respectively. Additionally, Figs. 6–9 show the success graphs of the ICE2DE-MDL prediction model on the individual stock datasets and stock market index. In these visualizations, the blue curve denotes the actual closing values of the indices/stocks, while the green curve depicts the denoised data acquired post the denoising technique application. The orange curve showcases the prediction outcomes derived from the ICE2DE-MDL model. When examining these figures, it becomes evident that the proposed denoising method effectively smoothens the abrupt fluctuations in the closing

**Table 5  Entropy values and ratios of IMFs acquired through the implementation of two decomposition stages for the KOSPI index.**

| | (i) Initial decomposition | | | | | (ii) Secondary decomposition | | | |
| --- | --- | --- | --- | --- | --- | --- | --- | --- | --- |
| | Approximate entropy | Sample entropy | Approximate entropy ratio | Sample entropy ratio | | Approximate entropy | Sample entropy ratio | Approximate entropy | Sample entropy ratio |
| *Imf-1"* | **0.424** | **3.683** | **18.386** | **29.404** | *Imf-1"* | **0.331** | **3.595** | **12.233** | **47.185** |
| *Imf-2'* | **0.470** | **3.438** | **20.399** | **27.451** | *Imf-2"* | **0.640** | **2.358** | **23.617** | **30.946** |
| *Imf-3(')* | 0.381 | 2.145 | 16.525 | 17.124 | *Imf-3"* | **0.715** | **0.754** | **26.400** | **9.899** |
| *Imf-4(')* | 0.217 | 1.267 | 9.439 | 10.119 | *Imf-4"* | **0.604** | **0.568** | **22.288** | **7.458** |
| *Imf-5(')* | 0.188 | 0.718 | 8.156 | 5.736 | *Imf-5(")* | 0.314 | 0.241 | 11.601 | 3.159 |
| *Imf-6(')* | 0.168 | 0.562 | 7.297 | 4.487 | *Imf-6(")* | 0.072 | 0.070 | 2.645 | 0.917 |
| *Imf-7(')* | 0.150 | 0.331 | 6.524 | 2.645 | *Imf-7(")* | 0.020 | 0.019 | 0.720 | 0.254 |
| *Imf-8(')* | 0.096 | 0.189 | 4.166 | 1.511 | *Imf-8(")* | 0.008 | 0.008 | 0.304 | 0.110 |
| *Imf-9(')* | 0.208 | 0.186 | 9.048 | 1.481 | *Imf-9(")* | 0.005 | 0.005 | 0.175 | 0.065 |
| *Imf-10(')* | 0.001 | 0.005 | 0.062 | 0.042 | *Imf-10(")* | 0.0004 | 0.001 | 0.016 | 0.008 |

**Notes.**
The bold values represent the IMP components classified as high-frequency.

**Table 6  Forecasting results for stock market indices.**

| Dataset | RMSE | MAE | MAPE | $R^2$ |
| --- | --- | --- | --- | --- |
| CAC 40 | 0.038 | 0.029 | 0.128 | 0.998 |
| FTSE 100 | 0.072 | 0.055 | 0.283 | 0.990 |
| KOSPI | 0.084 | 0.064 | 0.148 | 0.965 |
| NASDAQ 100 | 0.244 | 0.143 | 0.243 | 0.966 |
| NIKKEI | 0.164 | 0.123 | 0.484 | 0.919 |
| SET | 0.035 | 0.028 | 0.180 | 0.997 |
| S&P 500 | 0.117 | 0.066 | 0.165 | 0.990 |
| SSE | 0.031 | 0.026 | 0.314 | 0.995 |

**Table 7  Forecasting results for individual stocks.**

| Dataset | RMSE | MAE | MAPE | $R^2$ |
| --- | --- | --- | --- | --- |
| Xinning logistics | 0.044 | 0.035 | 0.238 | 0.988 |
| Zhongke electric | 0.221 | 0.144 | 0.594 | 0.905 |
| CITIC securities | 0.061 | 0.044 | 0.369 | 0.990 |

values of indices and stocks. The overall alignment between denoised data and prediction results underscores the model's ability to forecast price changes over time more accurately. This highlights the ICE2DE-MDL model's effectiveness in handling stock item noise and making more reliable predictions.

## Comparison with other models

To assess the effectiveness of the ICE2DE-MDL forecasting model, we conducted a comparative analysis with existing stock forecasting models detailed in the preceding section. The performance comparison results with four benchmark models were provided

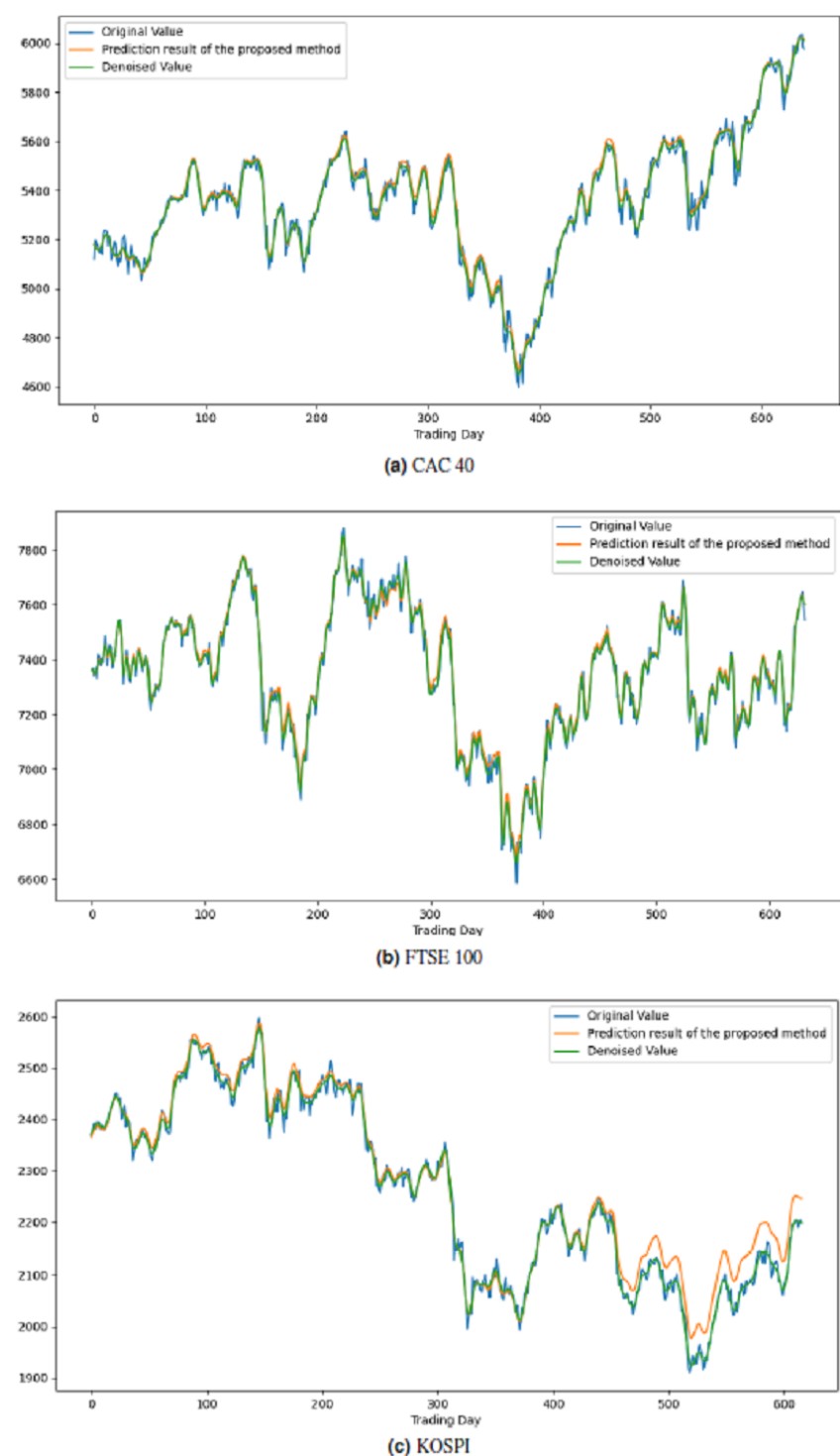

**Figure 6** Graph depicting the performance of the proposed prediction model on the test dataset of stock market indices.

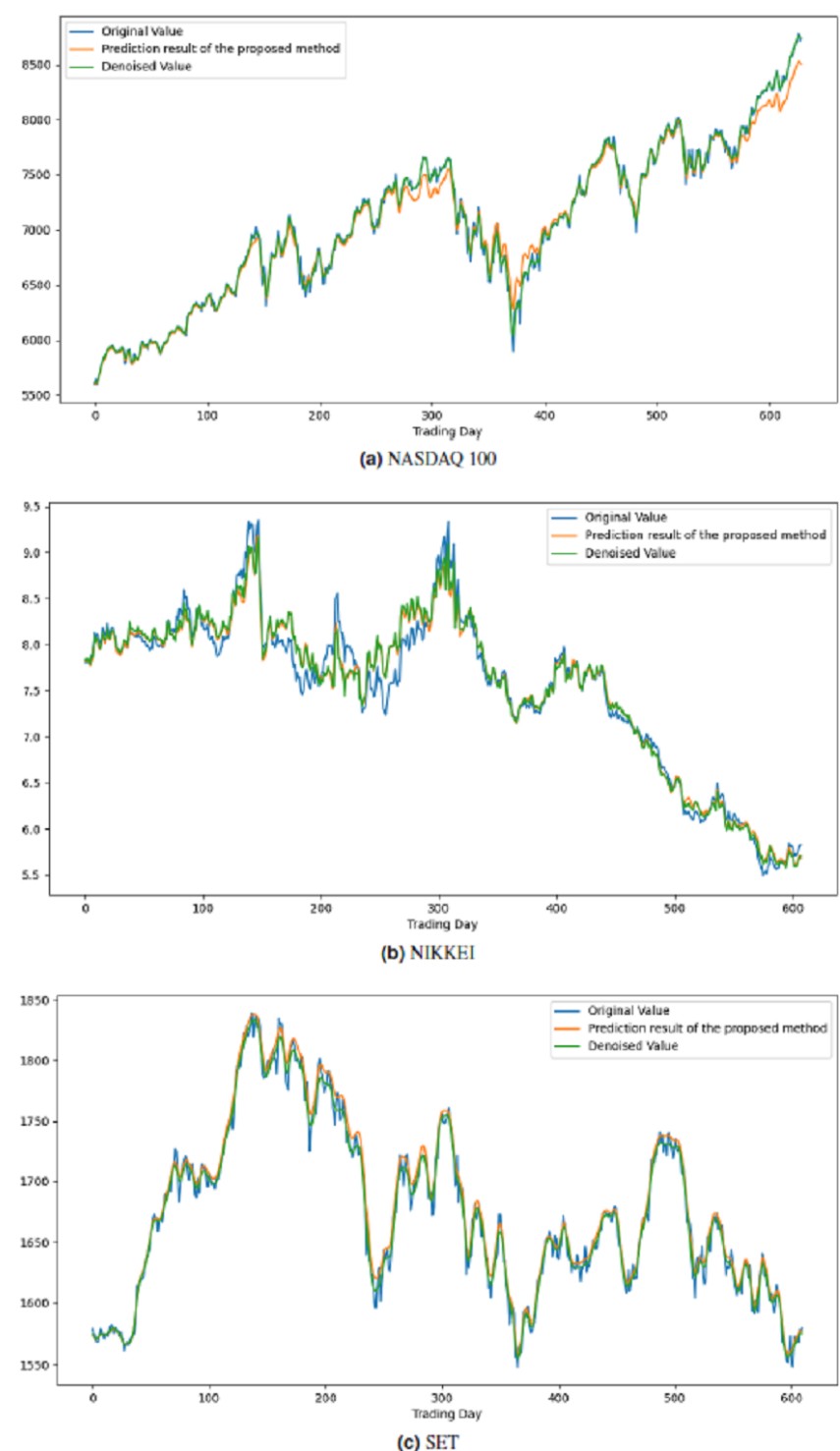

**(a) NASDAQ 100**

**(b) NIKKEI**

**(c) SET**

**Figure 7** Graph depicting the performance of the proposed prediction model on the test dataset of stock market indices.

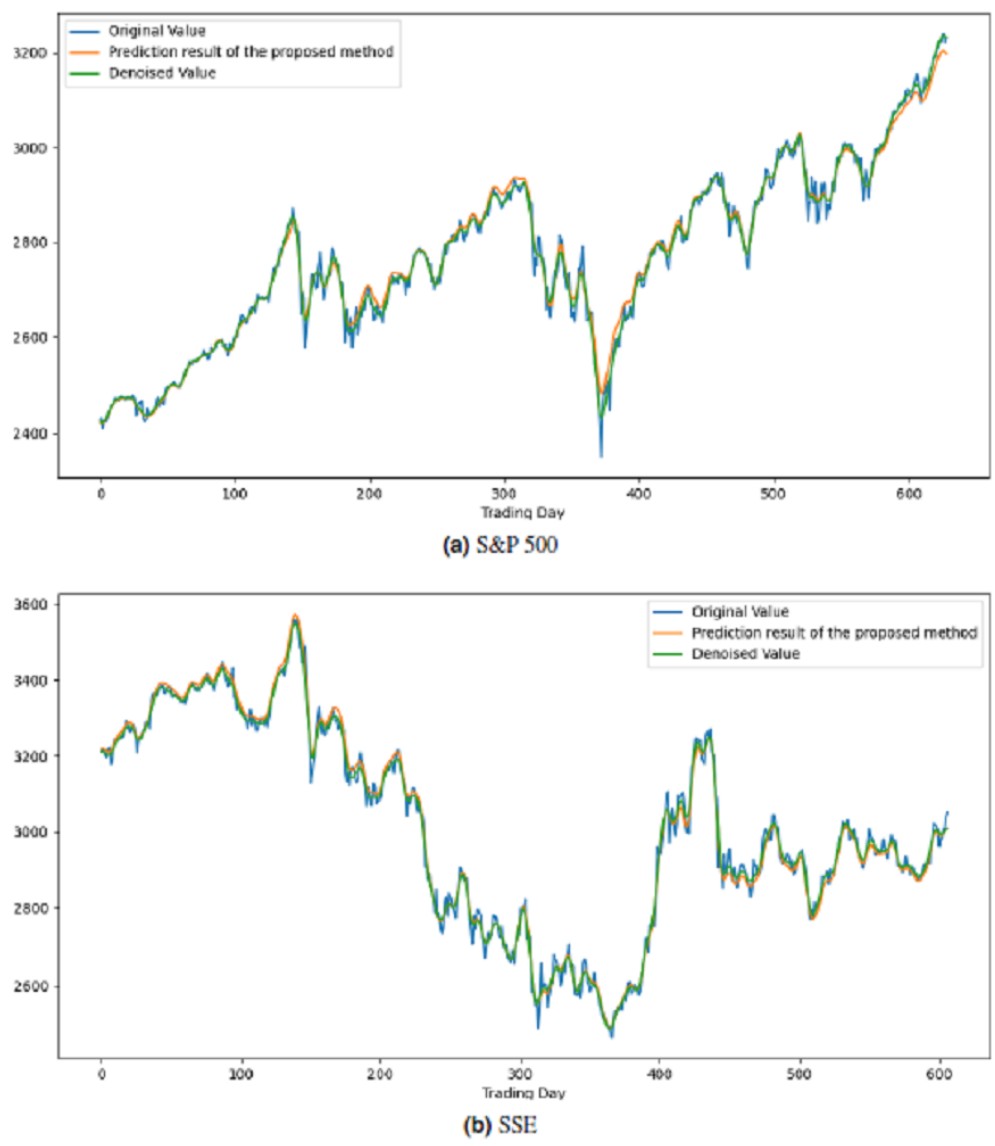

**Figure 8** Graph depicting the performance of the proposed prediction model on the test dataset of stock market indices.

in Tables 8–11. Additionally, numerical values highlighted in bold in the tables represent the best prediction results.

When comparing the ICE2DE-MDL prediction model with the VMD-SE-GBDT-BiGRU-XGBoost model, it is observed that our model exhibits lower MAE and RMSE metrics for all three stock market indices, indicating a higher level of accuracy in overall prediction performance. The slightly higher MAPE value may be attributed to a suboptimal hyperparameter tuning of the model. The higher MAPE value, stemming from hyperparameter tuning, suggests that there may be further improvement under specific

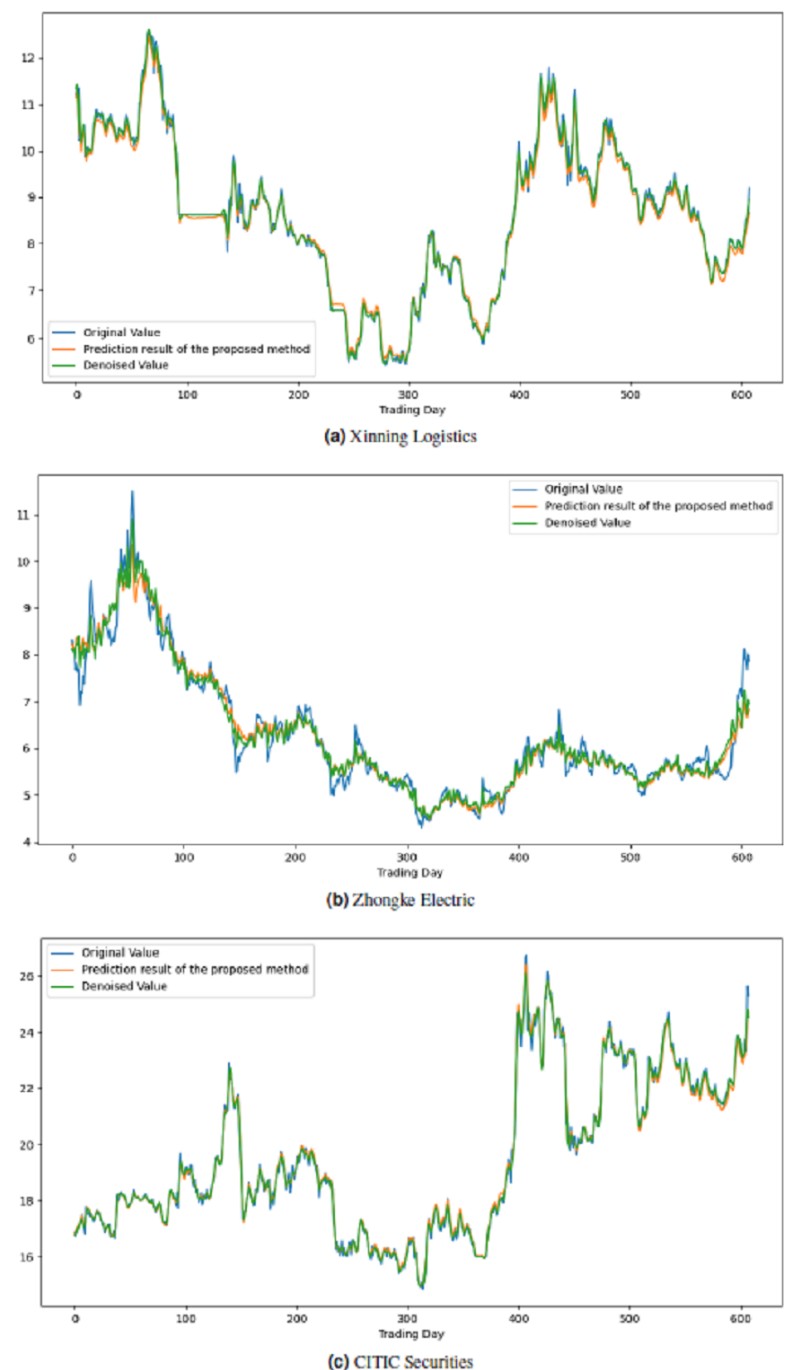

**Figure 9** Graph depicting the performance of the proposed prediction model on the test dataset of stock.

**Table 8   Performance comparison with VMD-SE-GBDT-BiGRU-XGBoost (*Wang, Cheng & Dong, 2023*) forecasting model.**

| Stock market index | Model | MAE | RMSE | MAPE |
|---|---|---|---|---|
| NASDAQ 100 | VMD-SE-GBDT-BiGRU-XGBoost | 130.074 | 174.479 | **0.014** |
| | ICE2DE-MDL (proposed model) | **0.143** | **0.244** | 0.243 |
| FTSE 100 | VMD-SE-GBDT-BiGRU-XGBoost | 74.128 | 102.056 | **0.011** |
| | ICE2DE-MDL (proposed model) | **0.055** | **0.072** | 0.283 |
| CAC 40 | VMD-SE-GBDT-BiGRU-XGBoost | 62.148 | 85.672 | **0.012** |
| | ICE2DE-MDL (proposed model) | **0.029** | **0.038** | 0.128 |

**Notes.**
Values highlighted in bold represent the best prediction results.

conditions. This indicates that the model's predictive capabilities could be enhanced in future studies through more effective hyperparameter tuning.

When compared to the IVMD-ICEEMDAN-ALSTM model, the ICE2DE-MDL prediction model exhibits the lowest values for all three error metrics, indicating a positive impact on model performance from both our denoising approach and the individual training of each IMF with an appropriate model. These results underscore the effectiveness of reducing noise in stock data and optimizing each IMF component separately, particularly enhancing the accuracy of the prediction model.

When compared to the sentiment analysis-based MS-SSA-LSTM prediction model and the P-FTD-RNN/LSTM/GRU model employing the padding-based Fourier transform denoising approach, the ICE2DE-MDL model has demonstrated superior performance across all stocks and stock market indices. This observation indicates that our suggested approach, as an alternative to sentiment analysis-based models and denoising methods, effectively enhances the overall prediction accuracy. The success of our model supports its robust and reliable forecasting capabilities under various conditions.

## CONCLUSION AND FUTURE WORKS

This study introduced a novel hybrid model, ICE2DE-MDL, integrating decomposition, entropy, machine, and deep learning techniques to improve stock prediction accuracy. ICE2DE-MDL was deployed to forecast the closing prices of eight stock indices and three individual stocks for the upcoming day. Performance assessments were conducted by comparing the ICE2DE-MDL with various hybrid models based on machine/deep learning methods in stock market predictions from the existing literature. These comparisons reveal that the ICE2DE-MDL outperformed current models on stock index and individual stock forecasting. A two-level denoising approach based on the ICEEMDAN effectively removed noise from financial time series without sacrificing valuable information. Notably, our experimentation also demonstrated that training IMFs with the most appropriate model, as opposed to a uniform model, significantly enhances overall model performance.

The ICE2DE-MDL model exhibited notable advancements in forecast accuracy when contrasted with alternative models. Nonetheless, opportunities for further refinement and optimization of this proposed hybrid approach persist. Future research endeavors could

**Table 9** Performance comparison with IVMD-ICEEMDAN-ALSTM (*Wang et al., 2022*) forecasting model.

| Stock market index | Model | MAE | RMSE | MAPE |
|---|---|---|---|---|
| SSE | IVMD-ICEEMDAN-ALSTM | 18.653 | 24.311 | 0.612 |
| | ICE2DE-MDL (proposed model) | **0.026** | **0.031** | **0.314** |
| NIKKEI | IVMD-ICEEMDAN-ALSTM | 196.414 | 267.002 | 0.903 |
| | ICE2DE-MDL (proposed model) | **0.123** | **0.164** | **0.484** |
| KOSPI | IVMD-ICEEMDAN-ALSTM | 13.941 | 17.309 | 0.606 |
| | ICE2DE-MDL (proposed model) | **0.064** | **0.084** | **0.148** |
| SET | IVMD-ICEEMDAN-ALSTM | 5.869 | 8.409 | 0.402 |
| | ICE2DE-MDL (proposed model) | **0.028** | **0.035** | **0.180** |

**Notes.**
Values highlighted in bold represent the best prediction results.

**Table 10** Performance comparison with MS-SSA-LSTM (*Mu et al., 2023*) forecasting model.

| Stock | Model | MAPE | RMSE | MAE | $R^2$ |
|---|---|---|---|---|---|
| Xinning Logistics | MS-SSA-LSTM | **0.033** | 0.258 | 0.178 | 0.958 |
| | ICE2DE-MDL (proposed model) | 0.238 | **0.044** | **0.035** | **0.988** |
| CITIC Securities | MS-SSA-LSTM | **0.025** | 0.790 | 0.647 | 0.941 |
| | ICE2DE-MDL (proposed model) | 0.369 | **0.061** | **0.044** | **0.990** |
| Zhongke electric | MS-SSA-LSTM | **0.042** | 1.345 | 0.957 | **0.979** |
| | ICE2DE-MDL (proposed model) | 0.594 | **0.221** | **0.144** | 0.905 |

**Notes.**
Values highlighted in bold represent the best prediction results.

**Table 11** Performance comparison with P-FTD-RNN/LSTM/GRU (*Song, Baek & Kim, 2021*) forecasting model.

| Stock market index | Model | MAE | RMSE | MAPE |
|---|---|---|---|---|
| S&P 500 | P-FTD-RNN | 12.822 | 19.577 | 0.469 |
| | P-FTD-LSTM | 9.252 | 15.855 | 0.344 |
| | P-FTD-GRU | 11.359 | 18.313 | 0.424 |
| | ICE2DE-MDL (proposed model) | **0.066** | **0.117** | **0.165** |
| SSE | P-FTD-RNN | 10.031 | 13.684 | 0.338 |
| | P-FTD-LSTM | 9.667 | 13.091 | 0.325 |
| | P-FTD-GRU | 10.903 | 14.395 | 0.365 |
| | ICE2DE-MDL (proposed model) | **0.026** | **0.031** | **0.314** |
| KOSPI | P-FTD-RNN | 9.915 | 12.857 | 0.449 |
| | P-FTD-LSTM | 7.795 | 10.669 | 0.362 |
| | P-FTD-GRU | 8.594 | 11.572 | 0.394 |
| | ICE2DE-MDL (proposed model) | **0.064** | **0.084** | **0.148** |

**Notes.**
Values highlighted in bold represent the best prediction results.

explore the impact of leveraging successful deep learning methods from various domains in stock prediction, aiming to elevate the overall performance of the prediction model.

Additionally, forecast accuracy can be increased by examining and including factors that affect stock prices rather than focusing only on closing prices.

### Funding
This work was supported by Ondokuz Mayıs University BAP under grant PYO.MUH.1904.23.002. The funders had no role in study design, data collection and analysis, decision to publish, or preparation of the manuscript.

### Grant Disclosures
The following grant information was disclosed by the authors:
Ondokuz Mayıs University BAP: PYO.MUH.1904.23.002.

### Competing Interests
The authors declare there are no competing interests.

### Author Contributions
- Zinnet Duygu Akşehir conceived and designed the experiments, performed the experiments, analyzed the data, performed the computation work, prepared figures and/or tables, authored or reviewed drafts of the article, and approved the final draft.
- Erdal Kılıç conceived and designed the experiments, analyzed the data, authored or reviewed drafts of the article, and approved the final draft.

### Data Availability
The data and code are available at GitHub and Zenodo:

- https://github.com/daksehir/ICE2DE-MDL-version-2/tree/version2.0

- daksehir. (2024). daksehir/ICE2DE-MDL-version-2: ICE2DE-MDL version-2 (version2.0). Zenodo. https://doi.org/10.5281/zenodo.11190408.

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
