# Peer review of "Multi level perspectives in stock price forecasting: ICE2DE-MDL"

_PeerJ Computer Science, doi:10.7717/peerj-cs.2125_

## Round 0.1 · original submission · Major Revisions

The authors need to clarify the motivations and contributions of this work. Furthermore, a detailed discussion is needed for the model and methods.

**Language Note:** The review process has identified that the English language must be improved. PeerJ can provide language editing services - please contact us at [email protected] for pricing (be sure to provide your manuscript number and title). Alternatively, you should make your own arrangements to improve the language quality and provide details in your response letter. – PeerJ Staff

·

Basic reporting

Article is well structured, and provides adequate significance, However, the article should go through some minor modification before proceeding any further.

1) The abstract needs to be provide some numerical findings.
2) Introduction can be elaborated to provide the overview of the experiment and its significance/contribution.
3)Some of the key terms in Algorithm 1 ICEEMDAN Decomposition Method can be elaborated for the reader.
4) In my opinion the rationale behind introducing ICE2DE-MDL prediction model, should be given in Research methodology section.
5) Figure 3. The flowchart of the proposed ICE2DE-MDL model can be more readable and clear.
6) Conclusion lacks the numerical comparison of introduced model with the existing models.

Experimental design

Well-defined

Validity of the findings

Adequate

Additional comments

None

Cite this review as

Reviewer 2 ·

Basic reporting

Proposed hybrid model aims to predict a stock closing price by integrating secondary decomposition, entropy, machine and deep learning methods. References to the literature is up to date and the paper is well organized. Experimental results also indicates the effectiveness of the method. various datasets are used to show the priority of the proposed approach. Research question is also well defined.

Experimental design

The authors give detailed results for the experiment. They evaluated the proposed model using commonly used metrics in the literature.The results also give a fair comparison with recently works. forecasting results are also given for stock market Indices and for individual stocks. Performance comparison has been realized with VMD-SE-GBDT-BiGRU-XGBoost and IVMD-ICEEMDAN-ALSTM, MS-SSA-LSTM. These works are also new for the field. Thus experimental results show the superiority of the method.

Validity of the findings

The paper can be considered for the publication. Conclusions are well stated and experiments support the novelty of the method.

Cite this review as

Reviewer 3 ·

Basic reporting

The outhors propose a hybrid model, ICE2DE-MDL, which aims to predict stock closing prices by integrating techniques from secondary decomposition, entropy analysis, and both machine and deep learning methodologies. The research highlights the development of a denoising method that utilizes entropy and the ICEEMDAN methodology to filter out noise from financial time series data, which is an innovative approach within the financial analytics sphere.

Experimental design

However, the manuscript would benefit significantly from a more detailed explanation of how each component of the model contributes to the overall prediction accuracy. The paper mentions the use of various deep learning and machine learning methods like LSTM, LSTM-BN, GRU, and SVR, applied to IMFs classified as noiseless, but it lacks an in-depth discussion of why these particular methods were chosen and how they interact within the model.

Validity of the findings

There are significant differences on the MAPE values between the results of the proposed model and two papers in the literature they compared given in Table 8 and 9. The authors should explain the reason. Also provide coding (python or matlab) and hardware details for the system used for experimentation. It would be great the if the authors upload the code of their experiments in a public area.

Cite this review as

---

## Round 0.2 · accepted · Accept

All the comments have been addressed. The paper is now acceptable for publication.

·

Basic reporting

All of my concerns are addressed. I have no further concerns for improving the manuscript. I suggest to accept this article.

Experimental design

no comments

Validity of the findings

no comments

Additional comments

no comments

Cite this review as

Reviewer 3 ·

Basic reporting

The paper can be accepted for publication in its present form with no new revisions.

Experimental design

The paper can be accepted for publication in its present form with no new revisions.

Validity of the findings

The paper can be accepted for publication in its present form with no new revisions.

Cite this review as